# Easy, Quick, and Reproducible Sonochemical Synthesis of CuO Nanoparticles

**DOI:** 10.3390/ma12050804

**Published:** 2019-03-08

**Authors:** Nataly Silva, Sara Ramírez, Isaac Díaz, Andreina Garcia, Natalia Hassan

**Affiliations:** 1Advanced Mining Technology Center (AMTC), Universidad de Chile, Av. Tupper 2007, Santiago 8370451, Chile; n.silvag@ug.uchile.cl; 2Department of Materials Chemistry, Faculty of Chemistry and Biology, Universidad de Santiago de Chile, Av. Libertador Bernardo O’Higgins 3363, Estación Central, Santiago 9170124, Chile; Sara.ramirez@usach.cl; 3Department of Chemical Engineering and Biotechnology, Faculty of Physical and Mathematical Sciences, Universidad de Chile, Beauchef 851, Santiago 8370456, Chile; isaac.diaz@ing.uchile.cl; 4Programa Institucional de Fomento a la I+D+i, Universidad Tecnológica Metropolitana. Ignacio Valdivieso 2409, San Joaquín, Santiago 8940577, Chile

**Keywords:** copper, nanoparticles, particles, synthesis, sonochemical, nanosized, X-ray techniques

## Abstract

Copper oxide nanoparticles (CuO NPs) were synthesized in air by reducing copper (II) sulfate pentahydrate salt (CuSO_4_·5H_2_O) in the presence of sodium borohydride. The reaction was stabilized with Hexadecyltrimethylammonium bromide (CTAB) in a basic medium and using ultrasound waves. Different molar ratios of CTAB:Cu^2+^ and NaBH_4_:Cu^2+^ were explored, to optimize the synthesis conditions, and to study the stability, size, and Zeta potential of the colloidal suspension. Optimum conditions to generate spherical, stable, and monodispersed nanoparticles with hydrodynamic diameters of 36 ± 1.3 nm were obtained, using 16 mM CTAB and 2 M NaBH_4_ (molar ratios Cu^2+^:CTAB:NaBH_4_ of 1:6:10). X-ray diffraction (XRD) was implemented, and a monoclinic CuO crystal system was formed. This demonstrated a monoclinic crystal system corresponding to CuO. The diffraction peaks were identified and confirmed according to their selected area electron diffraction (SAED) patterns.

## 1. Introduction

In the last few years, the synthesis of copper nanostructured materials, specifically copper nanoparticles (Cu NPs), cupric oxide nanoparticles (CuO NPs), and cuprous oxide nanoparticles (Cu_2_O NPs) has been of great interest, due to the attractive physical and chemical properties of these types of particles. Namely, particles of copper origin pose low toxicity to humans, are in high natural abundance, are low in cost to produce, and there are multiple ways to synthesize them [1,2,3,4]. 

CuO NPs are p-type oxide semiconductors with unique catalytic, optical, and electrical properties. The unique properties of these particles allow them to be used widely in applications in different fields, such as in catalysis [5], optoelectronics [6], as sensors [7], and as antimicrobial agents [8], among others [9]. 

The physicochemical characteristics of CuO NPs determine the types of application for which they can be used. For example, there is evidence that the types, shapes, and sizes of copper materials affect their toxicities. Some studies have found that NPs are more toxic than their micro-sized analogues, and these properties can influence the mechanisms conferring toxicity [10]. Moreover, Zakharova et al. have shown that particles of dimensions that are smaller than 50 nm spurn a greater antimicrobial response than do higher particles [11]. Further, Alzahrani et al. synthesized Cu NPs with different shapes and sizes, and evaluated their application as hydrogen sensors. They found that the spherical 37 nm Cu NPs best catalyzed the reduction of H_2_O_2_ [12]. Additionally, the reactivity of NPs also depends on the morphology, i.e., the crystal plane of the catalyst. Therefore, to achieve high photocatalytic performance, NPs with high surface-to-volume ratios are preferable [13,14]. 

Finally, nanoparticles in stable colloidal suspensions, and with specific zeta potentials are also conditioning factors for some applications. For example, it is known that the bioavailability of Cu, according to in vitro and in vivo biomedical experiments, determines the toxicities of NPs, and the overall toxicity is mainly influenced by three factors: the size, surface charge, and dissolution of the particles. Small, positively-charged nanoparticles in suspension confer high toxicities and they are well- distributed, thus facilitating interactions between cells and NPs [9].

For these and other reasons, it is valuable to study the optimal conditions that are needed to synthesize CuO nanostructures with specific sizes, shapes, charges, and stabilities. This being said, the simple, fast, and novel synthesis of CuO NPs with specific characteristics is a scientific challenge. 

Several methods to obtain CuO NPs have been proposed. For example, hydrothermal [15] and electrochemical [16] synthesis assisted by microwaves [17], pulsed laser evaporation [18], and sonochemical techniques [19,20,21,22,23,24] have been used to fabricate CuO NPs, and these methods greatly differ in the time that it takes to synthesize the particles. Moreover, the sizes and shapes of the NPs generated from these methods can differ.

Sonochemical synthesis has been used to efficiently produce CuO NPs of various different morphologies (spherical [24], rods [19], leaves [22], amorphous [22], discs [22], wires, and ellipsoidal [21]). Stabilizing agents such as polymers (polyvinyl alcohol (PVA) [23,25], polyethylene glycol (PEG), and polypropylene glycol (PPG) [23]), and surfactants (Hexadecyltrimethylammonium bromide (CTAB) [19,21,22], hexamethylene tetramine (HTMA) [19], and tetramethylammonium hydroxide (TMAH) [24]) are often added for morphological control. Despite this, most oxides obtained have sizes over 50 nm and are irregularly shaped. For the case of CuO NPs stabilized with polymers, the particles obtained after sonication are subjected to high temperatures to complete the synthesis. The calcined temperature is a crucial factor affecting the size and crystallization of NPs (the higher the temperature, the smaller the particle size) [23,25]. The morphologies of CuO NPs obtained using CTAB, on the other hand, depends on the amount of surfactant that is added. Specifically, low-level additions of CTAB allow for the formation of ellipsoidal nanostructures, whereas nanowires result when the amount of CTAB is high [21]. Other authors have reported that nanodiscs can be generated from the sonochemical synthesis of CuO using CTAB, whereas irregularly shaped flake-like structures are obtained when CTAB is absent [19]. Experiments with different concentrations of NaOH, ethanol, and CTAB lead to changes in the morphology and size of CuO nanoparticles [19]. Other authors have reported that the sonochemical synthesis of CuO using CTAB results in nanodiscs, whereas irregularly shaped flake-like structures are obtained without CTAB [22]. However, for the moment, there are no reports on the production of colloidal suspensions of CuO nanospheres (< 50nm) that have been stabilized with CTAB and obtained with sonochemical methods.

In general, several synthesis routes have been reported to obtain nanostructures that are stabilized with CTAB, and all of these suggest that it is possible to tune the shapes of nanomaterials by regulating the concentration of the surfactant. The cationic surfactant has a critical micellar concentration (CMC) of 0.9 mM [26]. When the synthesis conditions are [CTAB]< CMC, isolated monomers stabilize the nanoparticles. However, when [CTAB]>CMC, CTAB acts as a micellar template to give rise to various forms, such as spheres, rods, wires, and others [27,28,29,30,31]. Therefore, controlling the concentration of CTAB used can allow one to obtain the desired morphology. 

Various reducing agents have been used for the synthesis of copper nanomaterials. These reducing agents include: sodium citrate [32], reducing sugar [33], ascorbic acid [34], hydrazine [35], NaBH_4_ [36], and H_2_ [37], among others [3]. The physicochemical properties of nanoparticles, such as particle size, reactivity, and oxidation state can be controlled by multiple factors, including the type of reducing agent used, the concentration and the molar ratio of the metal ion reducing agent, the pH, the dripping rate, and the mixing method of the reactants [36]. For example, the use of hydrazine favors the formation of small metal particles, because the oxidation of the reducing agent generates N_2_ in the solution. This is equivalent to keeping the solution purged with inert gas; thus, oxidation can be avoided for prolonged periods [38]. Moreover, H_2_ gas has also been used for the reduction of Cu^2+^, and this results in the formation of Cu_2_O NPs and Cu NPs with sizes of between 30 and 120 nm. The sizes of the NPs formed are controlled by adjusting the concentration of the metal, and/or the reduction reaction conditions. H_2_ can only be used as a reductant at high temperatures and/or high pressures, and some catalysts have also been added to these reactions [37,39]. NaBH_4_ has also been used extensively as a reductant for the production of nanoscale metal and metal boride particles [40,41]. Lastly, borohydride has been used as a stabilizing agent of silver nanoparticles [42].

Thus, to our knowledge, the use of CTAB and NaBH_4_ in the sonochemical synthesis of spherical and monodispersed CuO NPs with hydrodynamic diameters of less than 50 nm and with positive surface charges has not yet been reported.

In this work, we describe methods for the simple, quick, and reproducible synthesis of CuO NPs from a metal copper cluster. Furthermore, the effects of the size, Zeta potential, and stability of the colloidal suspension were found to depend on the molar ratio between Cu^2+^ and the stabilizing and reduction agents. 

## 2. Materials and Methods

CuO NPs were synthesized in a basic aqueous solution, using CTAB as the stabilizing agent and NaBH_4_ as the reducing agent. All chemicals of analytical grade were purchased from Sigma-Aldrich, and were used as received. CuSO_4_·5H_2_O (98%) and NaOH (≥ 99%) were obtained from Merck. CTAB (≥ 99%) and NaBH_4_ (99.99%) were procured from (Sigma). 

The CuO NPs were produced by ultrasound-assisted chemical reduction (digital ultrasonic BIOBASE, model UC-20A, 40 KHz). Initially, in order to find the optimal synthesis conditions, the concentrations of NaOH (4.4 mM), CuSO_4_·5H_2_O (0.4 M) and NaBH_4_ (2 M) were fixed, and five concentrations of CTAB (2.7 mM, 8 mM, 16 mM, 21.3 mM, and 26.7 mM) were used to evaluate 1, 3, 6, 8 and 10 CTAB:Cu^2+^ molar ratios. After identifying the conditions that generated monodispersed CuO NPs, as confirmed by Dynamic Light Scattering (DLS) measurements, we used fixed concentrations of NaOH (4.4 mM), CuSO_4_·5H_2_O (0.4 M) and CTAB (16 mM) and seven concentrations of NaBH_4_ (0.2 M, 1 M, 2 M, 3 M, 10 M, and 20 M) to evaluate the molar ratios of NaBH_4_:Cu^2+^ (1, 5, 10, 15, 20, 50, and 100).

The synthesis of the nanoparticles using the optimal conditions is described. In the first step, 7.5 mL of NaOH (4.4 mM) and 7.5 mL of CTAB (16 mM) were added to a glass vial and homogenized in an ultrasonic bath at 30 °C for 5 min. Then, 50 µL CuSO_4_·5H_2_O (0.4 M) was added to the vial and homogenized for 5 min, obtaining a light blue copper hydroxide suspension. A cold and freshly prepared solution of 100 µL NaBH_4_ (2 M) was rapidly aggregated, and a marked change from light blue to dark brown was observed. The reaction was kept in an ultrasonic bath for 15 min, and then the vial was capped and stored at 23 °C. When the colloidal suspension turned yellow, the CuO NPs were obtained.

The synthesized NPs were characterized by using UV-Vis absorption spectroscopy with a LAMBDA 750 UV/Vis/NIR (Perkin Elmer, Waltham, MA, USA). Also, a Peltier Temperature Programmer (PTP), model PTP-1 (Perkin Elmer), was used at the set temperature of 25 °C. A Zetasizer Nano ZS90 ((Malvern, UK) was used to characterize the hydrodynamic diameters and Zeta potentials of the NPs. The samples were lyophilized prior to analysis by X-ray diffraction (Bruker D8, Billerica, MA, USA). To lyophilize the samples, they were frozen in a vial (15 mL) with liquid nitrogen, and then they were subjected to deep-vacuum conditions in a lyophilizer for 24 hr (FreeZone Freeze Dryers, Labconco, Kansas City, MO, USA). The crystalline structure of the lyophilized nanoparticles was characterized using X-ray diffraction. X-ray diffraction was carried out in a Bruker D8 Advance, Kα1 Cu (λ = 1.5406 nm). 

To determine the size, shape, and phases present in the product, High-resolution Transmission Electron Microscopy (HR-TEM) (FEI 2005) and selected area electron diffraction (SAED) analyses were carried out in a FEI 2005, model Tecnai ST F120, 80–120 kV (Hillsboro, OR, USA).

## 3. Results and Discussion

### The Sonochemical Method

A blue copper hydroxide suspension was obtained after adding Cu^2+^ to the basic CTAB solution. Then, NaBH_4_ was prepared in cold water, and bubbles were observed from the hydrolysis of water. Subsequently, when the NaBH_4_ solution was added to the vial containing Cu^2+^ and CTAB, abundant foam was formed, along with a color change from blue to dark brown. As is already known, NaBH_4_ is a potent reducing agent that has a standard reduction potential of −1.24 V in alkaline solutions [43], and the standard reduction potential of Cu^2+^/Cu^+^ is +0.15 V, and that of Cu^2+^/Cu^0^ is +0.34 V [44]. Therefore, NaBH_4_ can reduce Cu^2+^ ions to their lowest oxidation states (Cu^0^ or Cu^+^). NaBH_4_ is a powerful reducing agent, due to 1) the evolution of hydrogen [37,39,45]; 2) the ability for NaBH_4_ to directly reduce Cu^2+^ ions to Cu^0^ [36,38,46]; 3) due to its ability to directly reduce Cu^2+^ ions to Cu^0^.

It is very important to highlight that ultrasonic irradiation has the advantage of introducing a variety of physical and chemical effects derived from acoustic cavitation. In this work, the primary experimental condition that was used was the sonochemical method. Ultrasound can induce the formation of different metal oxide nanostructures, due to the following different reasons: (1) ultrasound can transform the shape of the micelles that are used as a template for the nanostructure formation [47,48]; (2) it accelerates the rearrangement of some reactions producing different molecules; (3) ultrasound has effects on the crystallization process [49,50]. The experimental conditions and hydrodynamic diameters, the polydispersity index (PDI), and the Zeta potential values are summarized in Table 1. By modifying the CTAB:Cu^2+^ molar ratios (1, 3, 6, 8 and 10) different stable colloidal suspensions were obtained. 

The CTAB concentrations that were used in this study were above the critical micellar concentration (CMC). Therefore, these molecules were organized in the aqueous solution as micelles. After the addition of copper salt, Cu^2+^ ions were distributed at the head group surface of the micelles due to the electrostatic attraction between the Cu^2+^ ions and the Br^−^ anions from CTAB molecules. Subsequently, the addition of NaBH_4_ reduced Cu^2+^ ions to Cu^0^, thus forming the Cu NPs. In previous studies, De et al. have reported the mechanism by which nanoparticles can be stabilized. The authors propose that NPs are formed in the periphery of the micelle where complete nanoparticle growth is not possible, due to the micellar steric effect. The NPs grow independently, forming spherical NPs that are polydispersed and limited in size [27]. On the other hand, Wu et al. explain that after reduction, a structural micellar rearrangement occurs, and the CTAB micelles cap the NPs after they are formed [35]. Our results demonstrate that at different CTAB concentrations above the CMC, the size of the NPs remain unchanged. This result reveals that CTAB acts as a stabilizer agent controlling the size of the Cu NPs, which is limited by the micelles formed. Therefore, we propose that all of the prepared samples arose via this mechanism. Higher concentrations of CTAB resulted in Zeta potentials that changed from 28 ± 1.2 mV to 42 ± 2.5 mV. The Zeta potential increased, due to the presence of quaternary ammonium salts. Additionally, seven colloidal suspensions were explored at different molar ratios of NaBH4:Cu^2+^ (1, 5, 10, 15, 20, 50 and 100) (Table 2). The colloidal suspension with a molar ratio of 1 turned yellow after the addition of NaBH_4_. From this, monodispersed NPs with hydrodynamic diameters of 84 ± 5.9 nm were formed. However, after 2 hr, the suspension became colorless and a yellow precipitate formed, indicating the presence of metastable copper hydroxide [51]. Given the molar ratios of 5 and 20, stable and monodispersed NPs were formed with hydrodynamic diameters of 50 nm. The obtained colloidal suspensions with molar ratios of 50 and 100, generated unstable nanoparticles with diameters of 79 ± 8.7 nm and 76 ± 6.5 nm, and the sizes of these particles increased until precipitation occurred after several hours. In addition, the samples turned yellow after three and four days, respectively. The time that it took to change the colloidal suspension color from dark brown to yellow was directly related to the amount of reducing agent that was added. This could be due to the fact that NaBH_4_ was still reacting, and the particles remained as dark brown metallic clusters. However, if the reducing agent is completely consumed, growth and oxidation of the particles occurs, forming copper oxide. The relationship between the size of the oxide nanoparticles and the concentration of NaBH_4_ can be explained in terms of the rate of reduction and the formation of metal nuclei.

According to the classical theory of nucleation, nanoparticles are formed in three steps: (1) pre-nucleation, (2) the formation of metal nuclei or clusters, and (3) growth. It has been shown that the higher the amount of reducer, the smaller the size of the metallic nanoparticles [35]. At a NaBH_4_:Cu^2+^ 1:1 molar ratio, we obtained large and unstable NPs over time, likely explained by a slow speed of reduction where only some nuclei were formed in the prenucleation step. For this reason, we propose that the reduction of Cu^2+^ that was formed in the later period is mainly used during the collision step once the nuclei is already formed, rather than during the formation of a new one. Therefore, this process led to the formation of larger particles. Indeed, this explanation is related to the formation of initial clusters, i.e. Cu NPs, corresponding to the dark brown solution. Then, as evaluated by DLS measurements, the CuO NPs, can be considered as a fourth step of nucleation by oxidation [52]. At higher concentrations of reducing agent, and with molar ratios between 5 and 20, the speed of reduction is faster than the nucleation process. Most of the Cu^2+^ ions are reduced prior to nuclei formation, generating nuclei that form small Cu NPs, and consequently small and monodispersed CuO NPs. Similar results have been previously obtained by using hydrazine as a reducer, where the resulting NPs have reproducible diameters when the concentration of the reducing agent increases [35]. However, when the NaBH_4_ concentrations is higher, i.e., at molar ratios between 50 and 100, another phenomenon of instability occurs, mainly because many highly reactive nuclei rapidly coalesce, forming particles with high PDI until they precipitate. Additionally, H_2_ can contribute to the reduction speed, because the temperature and pressure in sonochemical conditions are higher at the cavitation regions, favoring their utilization as reducers.

In the Zeta potential measurements, no tendencies was observed when we modified the concentration of NaBH_4_. This result confirms that the value of the Zeta potential is associated with the charge that the stabilizing agent provides on the NPs’ surface, independently of the particle size.

Stable and a monodisperse colloidal suspensions were obtained at 16 mM CTAB and 2 M NaBH_4_ (Cu^2+^:CTAB:NaBH_4_ molar ratios of 1:6:10). These suspensions were characterized by UV-Vis spectroscopy, and the absorption spectra were registered every 15 min for 20 hr. The UV–Vis spectra shown is plotted every four hours (Figure 1). 

The Cu NPs of the dark brown suspension showed a absorption band at 570 nm (a-band) that was attributed to the surface plasmon resonance that was reported between 500–600 nm [53]. Cu NPs are still stable after a period of 11 hr; afterwards the colloidal suspensions changed their colors from dark brown to yellow and a bathochromic shift, decreasing the signal intensity (shown in the insert of Figure 1). This color change corresponds to the oxidation of metallic copper, in agreement with previous reports indicating that copper oxide has an intense yellow color [54], with a unique absorption band of between 400–500 nm of significant intensity, corresponding to the oxide (c band) [55]. The b and c bands belong to O–Cu–O and Cu–O–Cu charge transfer bands [56], and after 11 hr, both bands increase in intensity. Other authors have reported that CuO NPs have two absorption bands at 220 nm and 350 nm [57]. 

To determinate the size, shape, and charge of the NP after each synthesis step, the dark brown and yellow suspensions were characterized by TEM, DLS, and the Zeta potential. The TEM micrographs (Figure 2a) revealed that the dark brown colloidal suspension corresponded to metal clusters of 3.6 ± 1.4 nm. By the DLS measurements, we obtained a distribution size number for Cu NPs of 2 ± 0.3 nm, and for the distribution size intensity there were two peaks of 80 ± 3.8 nm and 9 ± 1.3 nm, respectively. Interestingly, a high polydispersity index (PDI) value of 0.592 was obtained. The DLS measurements suggest that while NaBH_4_ continues to react, the colloidal suspension would be polydispersed. This could be explained by the presence of metallic nuclei, and micelles of dynamic sizes. That being said, it is important to highlight that the bubbles obtained by H_2_ formation impact the PDI value. Subsequently, a value of 39 ± 2.6 mV was obtained by the Zeta potential. Figure 2b shows the TEM micrographs that were obtained after 72 hr in the yellow colloidal suspension of copper oxide, in which spherical particles 34 ± 12 nm in diameter and homogeneously distributed, were observed. Similar results were observed with DLS, where a distribution size number of 36 ± 1.3 nm and a PDI of 0.150, with a Zeta potential value of +37 ± 1.5 mV, were attained. 

The distribution size value (36 ± 1.3 nm) obtained in the DLS was the most representative diameter of the dense electronically nanosphere obtained through TEM. This difference with the distribution size intensity value of 59 ± 2.7 nm can be explained by the hydration sphere of the nanoparticles formed by the CTAB, ions, and the water adsorbed on the surface. Therefore, only distribution size numbers will be shown.

The copper oxide suspension was characterized by DLS after three and seven days of synthesis, obtaining diameters of 36 ± 2.5 nm (0.133 PDI) and 45 ± 1.4 nm (0.213 PDI), respectively. Moreover, after one month, we observed that a small precipitate of light brown color began to appear; however, the DLS measurement of the colloidal suspension showed a value of 50 ± 3.1 (0.253 PDI).

In order to determine the crystalline structures of the product, the lyophilized nanoparticles were characterized by X-ray diffraction. This procedure was carried out in a Bruker D8 Advance, Kα1 Cu (λ = 1.5406 nm). For comparative purposes, the diffractograms shown in Figure 3a represent the diffraction patterns that are obtained for lyophilized CuO NPs with CTAB and pure CTAB. The diffraction pattern obtained for the CuO NPs exhibited four different signals, which are not found in the pure CTAB diffractogram. These signals were indexed in the monoclinic crystal system, and they probably correspond to CuO diffraction peaks (JCPDS 00-048-1548) [58]. Other copper phases (cubic Cu_2_O) were not found in the diffractogram. Additionally, two important peaks appeared between 2θ of 30° and 33°. A peak observed at ca. 32.5 could correspond to the plane (110); however, its intensity should be lower (see Table 2) and this peak could be separated between the CTAB signal and CuO. The low intensity of the CuO peaks with respect to CTAB makes the XRD spectrum difficult to analyze. The closest peak to 30° could not be indexed to confirm whether this signal was due to CTAB, or whether it was from other copper phases. Therefore, we suggest that a more precise study should be carried out through electronic diffraction. 

To accurately study the crystalline structure of the synthesized copper oxide product, a selected area electron diffraction (SAED) was carried out for CuO NPs. The SAED pattern revealed in Figure 3b showed a polycrystalline structure with defined rings. This pattern was successfully indexed in the monoclinic system, and the interplanar distances calculated (d_hkl_) matched with those that were found in the CuO database [58]. The database used for the indexation was obtained from the powder diffraction files, explaining why many differences in the d_hkl_ distances were found in the calculation procedure. These values are recorded in Table 2. An HR-TEM micrograph demonstrates that the exposed atomic plane on the surface of the CuO NPs corresponds to (1 1 0), in agreement with previous reports of CuO NPs synthesis (Figure 3c) [59]. Subsequently, to determine the atomic directions that are present in HR-TEM, a Fast Fourier Transform (FFT) filter was applied. The FFT micrograph indicated the presence of the (1 1 0) and (1 1 -1) planes, following the CuO phase (Figure 3d). 

The XRD, together with SAED and HRTEM, allowed us to determine that the synthesized compound corresponded to copper oxide (II) (CuO). The mechanisms and representative schemes of the synthesis process of CuO NPs are shown below, and in Figure 4:(1)$$\mathrm{CTAB}+2{\mathrm{OH}}^{-}\overset{{\mathrm{Cu}}^{2+}}{\to }\mathrm{Cu}{\left(\mathrm{OH}\right)}_{2}-\mathrm{CTAB}$$
(2)$${\mathrm{NaBH}}_{4}+H_{2}O\to {\mathrm{Na}}^{+}+{\mathrm{OH}}^{-}+H_{3}{\mathrm{BO}}_{3}+4H_{2}$$
(3a)$$\mathrm{Cu}{\left(\mathrm{OH}\right)}_{2}-\mathrm{CTAB}+H_{2}\to {\mathrm{Cu}}^{0}-\mathrm{CTAB}+2H_{2}O\overset{11h}{\to }\mathrm{CuO}-\mathrm{CTAB}$$
(3b)$$4\mathrm{Cu}{\left(\mathrm{OH}\right)}_{2}-\mathrm{CTAB}+{\mathrm{BH}}_{4}^{-}\to 4{\mathrm{Cu}}^{0}-\mathrm{CTAB}+B{\left(\mathrm{OH}\right)}_{4}^{-}+6H_{2}O\overset{11h}{\to }\mathrm{CuO}-\mathrm{CTAB}$$

## 4. Conclusions

CuO NPs have been successfully obtained by using an easy, quick, and reproducible method of sonochemical synthesis. Unlike other methods, the synthesis does not require high temperatures or high pressures. The synthesis is mainly evidenced by two color changes: The first one is a dark brown suspension corresponding to copper metal nuclei, and the second one is a yellow color corresponding to copper oxide. The Zeta potential of the CuO NPs can be tuned by varying the CTAB concentration. Additionally, colloidal suspension stability and the CuO NP sizes are guided by the molar ratio of NaBH_4_:Cu^2+^. The optimum conditions were obtained with 16 mM CTAB and 2 M NaBH_4_ (Cu^2+^:CTAB:NaBH_4_ molar ratios of 1:6:10), generating spherical, stable, and monodisperse nanoparticles with hydrodynamic diameters of 36 ± 1.3 nm. The NPs are maintained in a stable colloidal suspension for up to one month. A monoclinic crystal system corresponding to CuO was evidenced by the patterns of DRX and SAED measurements.

## Figures and Tables

**Figure 1 materials-12-00804-f001:**
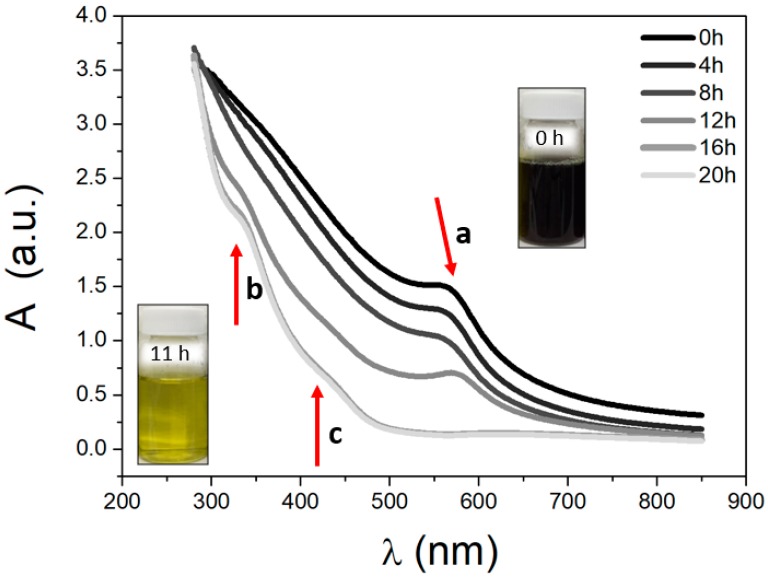
Evolution UV-Vis spectra of the colloidal suspensions recorded over 20 hr. Spectra were recorded at 20 °C between 300 and 850 nm, measured every 15 min. The insert shows a photograph of a glass vial that is similar to the colored colloidal suspension.

**Figure 2 materials-12-00804-f002:**
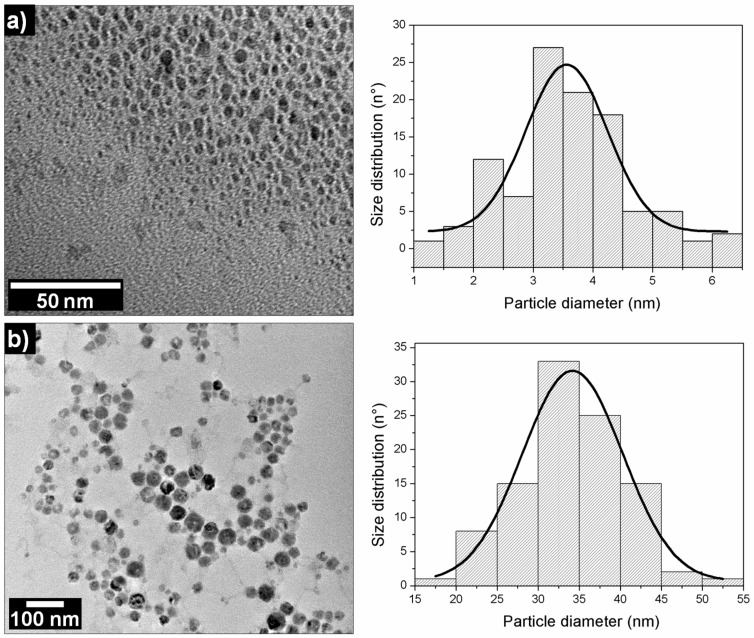
Transmission Electron Microscopy (TEM) micrographs of (**a**) copper nanoparticle (Cu NPs) and (**b**) copper oxide nanoparticle (CuO NPs) with their respective particle size distribution histograms.

**Figure 3 materials-12-00804-f003:**
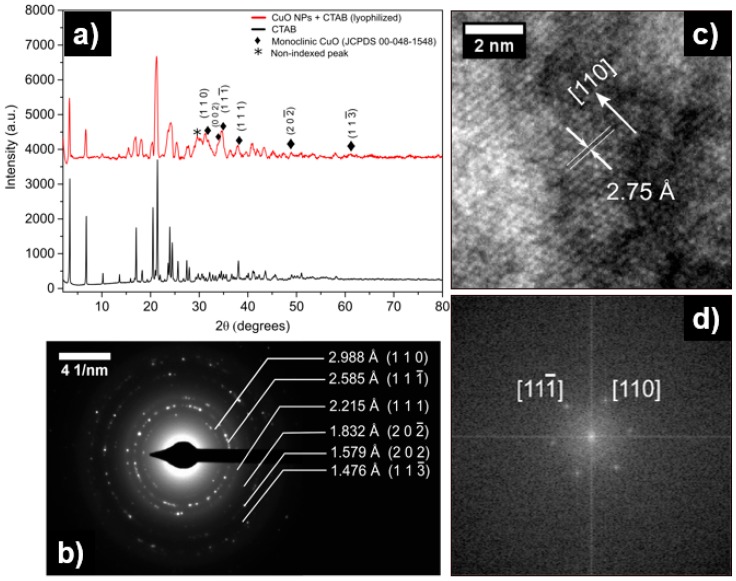
(**a**) X-ray powder diffraction patterns of CuO NPs and CTAB. (**b**) Selected areas of electron diffraction of CuO NPs. (**c**) HR-TEM and (**d**) Fast Fourier Transform (FFT) of CuO NPs.

**Figure 4 materials-12-00804-f004:**
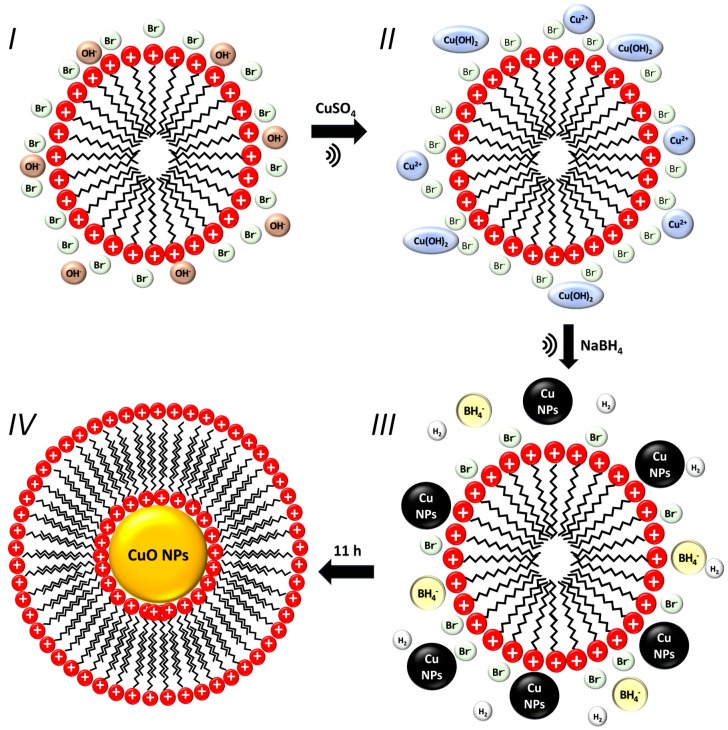
Schematic representation of the synthesis mechanism of CuO NPs obtained by the sonochemical method. Step I: CTAB micelle in basic solution (uncolored). Step II: Suspension of copper hydroxide (light -blue coloration). Step III: Reduction of Cu(OH)_2_ and Cu^2+^ by the synergic effects of NaBH_4_ and H_2,_ generating nuclei that contain small Cu NPs formed in the periphery of the micelle (dark brown suspension). Step IV: Nucleation by the oxidation of Cu NPs, and structural micellar rearrangement. The CTAB micelles cap the CuO NPs after they are formed (yellow colloidal suspension).

**Table 1 materials-12-00804-t001:** Hydrodynamic diameter, polydispersity index, and Zeta potential, determined by Dynamic Light Scattering (DLS) measurements of samples that are prepared at different molar ratios of Cu^2+^:CTAB:NaBH_4_.

Cu^2+^	CTAB	NaBH_4_	Diameter (nm)	PDI	Zeta Potential (mV)
1	1	10	42 ± 9.3	0.401	29 ± 2.5
1	3	10	35 ± 1.3	0.542	32 ± 3.1
1	6	10	36 ± 1.3	0.150	37 ± 1.5
1	8	10	36 ± 2.9	0.181	39 ± 1.7
1	10	10	38 ± 1.9	0.264	42 ± 2.0
1	6	1	84 ± 5.9	0.141	39± 0.8
1	6	5	30 ± 1.2	0.184	31 ± 1.0
1	6	10	36 ± 1.3	0.150	37 ± 1.5
1	6	15	34 ± 3.1	0.299	36 ± 1.7
1	6	20	40 ± 0.9	0.224	39 ± 2.0
1	6	50	79 ± 8.7	0.204	44 ± 1.8
1	6	100	77 ± 6.5	0.349	37 ± 3.2

**Table 2 materials-12-00804-t002:** Interplanar distances and intensity signals for different atomic planes in CuO (JCPDS 00-048-1548) [58].

Miller Index (h k l)	Interplanar Distance d_hkl_ (Å)	Intensity of Signal (%)
(1 1 0)	2.75201	13
(1 1 −1)	2.52367	100
(1 1 1)	2.32429	99
(2 0 -2)	1.86764	30
(2 0 2)	1.58227	10
(1 1 -3)	1.50660	20

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
