# Peer review of "Easy, Quick, and Reproducible Sonochemical Synthesis of CuO Nanoparticles"

_materials, 2019, doi:10.3390/ma12050804_

Reviewer 1 Report

The manuscript describes a sonochemical procedure for the fabrication of copper oxide nanoparticles (CuO NPs) via the reduction of CuSO4with sodium borohydride (NaBH4) using CTAB as stabilizer. Authors have investigated the influence of the amount of reducing agent and stabilizer on the size, zeta potential and stability of the CuO NPs solution. Although the topic -synthesis of CuO NPs- is of interest, the main drawback of the manuscript is that the novelty of synthetic procedure has not been properly justified. There are other papers in which CuO NPs have been prepared by reduction with NaBH4, which have not been included within the references. In addition, the manuscript could be improved if the following comments are taken into account: 

1)   “First name” should be removed from the list of authors. 

2)   Super and subscripts along the manuscript should be corrected (e.g. lines 18, 19, 20…).

3)  Introduction is too short. It should describe previous reported methods for the preparation of CuONPs, their limitations, as well as the advantages of the proposed method as compared those. The relevance and novelty of the synthetic procedure proposed should be emphasized. In addition, interest and potential applications of the prepared CuO NPs should be included in more detail. 

4)   There are related references regarding sonochemical synthesis of CuO NPs which are not mentioned in the manuscript, e.g.: Wongpisutpaisan et al. Energy Procedia 9 (2011) 404-409; Raujbar-Karimir et al. Physica B: Cond. Matter. 405 (2010) 3096-3100; Kumar et al. Langmuir 17 (2001) 1406-1410; Zhang et al. RSC Adv. 5 (2015) 28786-28793. 

5)   The synthesis of Cu nanoparticles with NaBH4has been previously described (references which are not included within the manuscript), e.g. Liu et al. Transactions of nonferrous metals society of China 22 (2012) 117-123; Suramwar et al. Arabian J. Chem. 9 (2016) 51807-51812; Litmanovich et al. Polymer Sci. Series B 53 (2011) 202. Thus, the novelty of the paper should be justified. 

6)   In addition, the review from Gawande et al. in Chem. Rev. 116 (2016) 3722-3811 states: “The wet chemical technique of a long-established approach for the preparation of metallic Cu NPs mostly involving reducing agents (e.g. sodium borohydride) that provide electrons for the reduction of Cu salts (e.g. CuSO4). Various capping agents have also been used to stabilize the ensuing CuNPs and to control the particle growth”. 

7)   Line 74: there is an extra “UV-vis”

8)   Line 75: please revise the sentence, it should be better clarified. 

9)   Why upon time and oxidation the size of the CuO NPs increases from approx. 3.6 nm to 34.18 nm? This effect should be discussed within the text. Similar increase in size measured by DLS is observed by increasing the number of days from synthesis.

10) Lyophilized NPs were used for characterization. The lyophilization conditions should be described in section 2. 

11) The study of the influence of variables should be discussed at the begging of section 3, instead of showing first the complete characterization at the optimal conditions. The different studies prior selecting such conditions should be described in advance. 

12) The mechanism of the synthesis is not clear. Do authors meant that hydrogen is the responsible of the reduction of Cu2+? Experimental evidences supporting the proposed mechanism should be included as well as bibliographic references. Other authors have reported different mechanisms for the reduction of Cu2+ to CuO NPs through NaBH4reduction. 

13) Lines 156-158: “No significant trend is observed in the Z-pot values because the variation is associated with the error of the measurements”. This sentence should be clarified. Does it mean that it is of the same range than that of the standard deviation of the measurements? 

14) Conclusions are too short and do not comprise the different aspects evaluated in the manuscript. 

15) Author contributions and Funding are not completed. 

Author Response

Referee 1

Comments and Suggestions for Authors

The manuscript describes a sonochemical procedure for the fabrication of copper oxide nanoparticles (CuO NPs) via the reduction of CuSO4with sodium borohydride (NaBH4) using CTAB as stabilizer. Authors have investigated the influence of the amount of reducing agent and stabilizer on the size, zeta potential and stability of the CuO NPs solution. Although the topic -synthesis of CuO NPs- is of interest, the main drawback of the manuscript is that the novelty of synthetic procedure has not been properly justified. There are other papers in which CuO NPs have been prepared by reduction with NaBH4, which have not been included within the references. In addition, the manuscript could be improved if the following comments are taken into account: 

1)  “First name” should be removed from the list of authors.

We apologize for the confusion, but we wrote the names as the journal format. In the general considerations for the manuscript preparation, points out: “Author List and Affiliations: Authors' full first and last names must be provided”.

2)  Super and subscripts along the manuscript should be corrected (e.g. lines 18, 19, 20…).

Thank you for the comment. The manuscript has been revised and modified.

3)  Introduction is too short. It should describe previous reported methods for the preparation of CuONPs, their limitations, as well as the advantages of the proposed method as compared those. The relevance and novelty of the synthetic procedure proposed should be emphasized. In addition, interest and potential applications of the prepared CuO NPs should be included in more detail.

Thank you for the comment. The introduction has been extended and we have incorporated the suggestions.

4)  There are related references regarding sonochemical synthesis of CuO NPs which are not mentioned in the manuscript, e.g.: Wongpisutpaisan et al. Energy Procedia 9 (2011) 404-409; Raujbar-Karimir et al. Physica B: Cond. Matter. 405 (2010) 3096-3100; Kumar et al. Langmuir 17 (2001) 1406-1410; Zhang et al. RSC Adv. 5 (2015) 28786-28793.

Thank you for the comment. The references were incorporated in the lines 59-60 and 63-68.

5) The synthesis of Cu nanoparticles with NaBH4 has been previously described (references which are not included within the manuscript), e.g. Liu et al. Transactions of nonferrous metals society of China 22 (2012) 117-123; Suramwar et al. Arabian J. Chem. 9 (2016) 51807-51812; Litmanovich et al. Polymer Sci. Series B 53 (2011) 202. Thus, the novelty of the paper should be justified.

Thank you for the comment. All references were incorporated in the lines 89, 95 and 155 and the novelty was re-confirmed at the introduction.

6)  In addition, the review from Gawande et al. in Chem. Rev. 116 (2016) 3722-3811 states: “The wet chemical technique of a long-established approach for the preparation of metallic Cu NPs mostly involving reducing agents (e.g. sodium borohydride) that provide electrons for the reduction of Cu salts (e.g. CuSO4). Various capping agents have also been used to stabilize the ensuing CuNPs and to control the particle growth”. The introduction has been extended and we have incorporated this reference and the different reducing agent and capping agents used in sonochemical synthesis.

7)   Line 74: there is an extra “UV-vis”

We apologize for the mistake, the extra “UV-vis” has been eliminated.

8)   Line 75: please revise the sentence, it should be better clarified.

We apologize for the unclear sentence, it has been modified and corrected in Lines 223-225.

9)   Why upon time and oxidation the size of the CuO NPs increases from approx. 3.6 nm to 34.18 nm? This effect should be discussed within the text. Similar increase in size measured by DLS is observed by increasing the number of days from synthesis.

Thank you for the comment. A better explanation of the stabilization and reduction mechanism was detailed between lines 195-204.  Discussion was focused in the modified parameters, such as the concentration of the reducing agent and the effect on the size, charge and stability of the CuO NPs.

10) Lyophilized NPs were used for characterization. The lyophilization conditions should be described in section 2. The lyophilization conditions were describe in lines 139-141 at the section 2. 

11) The study of the influence of variables should be discussed at the begging of section 3, instead of showing first the complete characterization at the optimal conditions. The different studies prior selecting such conditions should be described in advance.

Thank you for the comment. The paragraphs that describe the study of the different molar ratios of CTAB:Cu+2 and NaBH4:Cu+2 to optimize the synthesis were explained at the beginning of section 3, previous to the characterizations of the optimal conditions. Between lines 160-217.

12) The mechanism of the synthesis is not clear. Do authors meant that hydrogen is the responsible of the reduction of Cu2+? Experimental evidences supporting the proposed mechanism should be included as well as bibliographic references. Other authors have reported different mechanisms for the reduction of Cu2+ to CuO NPs through NaBH4reduction.

Thank you for the comment. A new paragraph has been incorporated to clarify the reduction process (Lines 145-159). Additionally, the mechanism has been modified and a schematic representation has been incorporated in figure 4.

13) Lines 156-158: “No significant trend is observed in the Z-pot values because the variation is associated with the error of the measurements”. This sentence should be clarified. Does it mean that it is of the same range than that of the standard deviation of the measurements?  

Thank you for the comment. The phrase was misunderstood; therefore, it has been modified. The following paragraph was added in Lines 214-217.

14) Conclusions are too short and do not comprise the different aspects evaluated in the manuscript.

The conclusions have been extended. Lines 307-317.

15) Author contributions and Funding are not completed.

The contributions and funding have been completed. Lines 319-323 and 324-325.

Reviewer 2 Report

The manuscript by Silva et al. deals with the sonochemical synthesis of CuO NPs via a novel approach where copper ions are reduced by sodium borohydride under ultrasounds and are stabilized by cetyltrimethylammonium bromide producing stable and monodisperse NPs. It is an interesting work but before publication a few issues have to be addressed and specially the interpretation of the XRD pattern as we see additional peaks and in advance the peaks assigned to the specific CuO JCPDS card are shifted. I have the following comments:

Correct all subscripts and superscripts in Abstract.

Replace throughout the manuscript CuONPs with CuO NPs.

Lines 56-57: what do you mean 7.5 mL of CTAB to different concentration?

Lines 72-73: as it is written, it seems that a solution of 16 mM CTAB and 2 M NaBH4 has molar ratios towards Cu2+ of 6 and 10 respectively, which can’t happen simultaneously. I can’t understand what you mean.

Figure 1 contains too many spectra (4*11=44?) which makes it hard to see specially in the region of 550 to 600 nm. You should present spectra every 1 or probably better every 2 hours and have this figure as SI. Furthermore, readers should know which colour (and spectrum) is which, and this would be better if you presented max 7 spectra (i.e. 0,2,4,6,8,10, and 11 h) and not even 12 (for every hour). In addition, in the caption you mention 21 h instead of 11 h.

Line 98: 34.18±12 nm is not correct. You should write 34±12 nm.

Line 100: same issue as above, you should write 36.0±1.3 nm.

Figure 3a: What about the peaks at ca. 30-32 degrees? Interpretation of the XRD pattern doesn’t seem good.

Lines 155-156: same issues with numbers as above.

Table 2: same issue.

Author Response

POINT TO POINT ANSWER TO QUESTIONS

Review 2

The manuscript by Silva et al. deals with the sonochemical synthesis of CuO NPs via a novel approach where copper ions are reduced by sodium borohydride under ultrasounds and are stabilized by cetyltrimethylammonium bromide producing stable and monodisperse NPs. It is an interesting work but before publication a few issues have to be addressed and specially the interpretation of the XRD pattern as we see additional peaks and in advance the peaks assigned to the specific CuO JCPDS card are shifted. I have the following comments:

Thank you for the comment. Regarding your first observation about the change in the position of the peaks with respect to the positions reported in PDF JCPDS 00-048-1548, we affirm that it is correct. The most noticeable displacement is observed in the peaks of planes 002 and 11-1, whose positions are 34.8° and 35°, respectively. The positions assigned in PDF card JCPDS 00-048-1548 for these planes correspond to 35.4 and 35.5 for planes 002 and 11-1, respectively. Other position changes of smaller magnitude are presented in the peaks of planes 111 and 20-2. These displacements may be due to stresses present in the CuO crystals generated during the crystal reduction and growth process.

Correct all subscripts and superscripts in Abstract.

Thank you for the comment. The manuscript has been revised and modified.

Replace throughout the manuscript CuONPs with CuO NPs.

All the abbreviations have been modified.

Lines 56-57: what do you mean 7.5 mL of CTAB to different concentration?

This line was eliminated. A new paragraph was added in the experimental methodology, to explain the parameters that were modified to find the optimal conditions of the synthesis. Subsequently, the synthesis with the optimized data was explained. Lines 117-125.

Lines 72-73: as it is written, it seems that a solution of 16 mM CTAB and 2 M NaBH4 has molar ratios towards Cu2+ of 6 and 10 respectively, which can’t happen simultaneously. I can’t understand what you mean.

The sentence has been modified to clarify the information. Lines 222-224.

Figure 1 contains too many spectra (4*11=44?) which makes it hard to see specially in the region of 550 to 600 nm. You should present spectra every 1 or probably better every 2 hours and have this figure as SI. Furthermore, readers should know which colour (and spectrum) is which, and this would be better if you presented max 7 spectra (i.e. 0,2,4,6,8,10, and 11 h) and not even 12 (for every hour). In addition, in the caption you mention 21 h instead of 11 h.

The Figure 1 has been modified and only the UV-Vis spectra measured every 4 hours were plotted.

Line 98: 34.18±12 nm is not correct. You should write 34±12 nm.

Decimals have been eliminated

Line 100: same issue as above, you should write 36.0±1.3 nm.

Decimals have been eliminated

Figure 3a: What about the peaks at ca. 30-32 degrees? Interpretation of the XRD pattern doesn’t seem good.

Thank you for the comment. We agree with your observation, between 30° and 33° there are two wide peaks with important intensity, which were not indexed in figure 3.a. In this figure it was only decided to include those peaks with an intensity above 20% (table 2) reason why peaks of planes (110) (2q = 32.5°) and (202) (2 q = 58.3°) were not included. The peak that appears ca.  2q = 32° can be related to the plane (110), however this should be decomposed between the signals of CTAB and CuO, task not easy to realize in those levels of intensity of signal. Comparing the pure CTAB signal respect to the CuO NPS + CTAB signal, it is observed that the signals corresponding to CTAB undergo a broadening when they are associated to CuO Nps. The XRD analysis presented in Figure 3.a was only carried out as an initial analysis to determine the presence of other copper phases. Peaks between 30º and 33º do not correspond to Cu and Cu2O. Because the CTAB signals hide the CuO peaks. More precise crystal structure analysis was carried out through electronic diffraction and TEM image analysis, which proved the formation of CuO and no other different phases were found.

A comment was added to clarify some of the limitations presented in the XRD analysis. Lines 273-278. In addition, CuO NPs was replaced by CuO NPs in figure 3.a.

Lines 155-156: same issues with numbers as above.

Decimals have been eliminated

Table 2: same issue.

Decimals have been eliminated

Reviewer 3 Report

The authors reported the preparation of CuO nanoparticles by sonochemical  synthesis. The results are not new. The authors simply reported the results without a real discussion. Several typos are present, even in the list of the authors. I do not recommend the publication of this paper

-The introduction is very short. The sonochemical preparation of CuO nanoparticles using CTAB as stabilizer has been already reported in the literature: Xiao et al, J Nanomaterials, 2011, doi.org/10.1155/2011/439162 CuO CTAB sonochemical; Purnachander et al, Energy and Environment FocusVol. 4, pp. 250–255, 2015 CuO CTAB sonochemical; Kim et al Ceramics International 42 (2016) 19454–19460 CuO CTAB sonochemical; Karunakaran, Journal of Alloys and Compounds 580 (2013) 570–577.The papers must be cited in the introduction.

-The authors claim that the novelty of this paper is the utilization of NaBH4. However it is not clear the advantage compared to the classical method (see article reported above)

-In general the quality of the images is poor. For example,in  Figure 1 too many spectra are present. and the results are not clear

-XRD  pattern, TEM images and particle distribution of all the prepared materials should be reported in the main text.

-The authors should discuss the influence of CTAB and NaBH4 on the  final particle size.

Author Response

POINT TO POINT ANSWER TO QUESTIONS

Referee 3

The authors reported the preparation of CuO nanoparticles by sonochemical synthesis. The results are not new. The authors simply reported the results without a real discussion. Several typos are present, even in the list of the authors. I do not recommend the publication of this paper

 The introduction is very short. The sonochemical preparation of CuO nanoparticles using CTAB as stabilizer has been already reported in the literature: Xiao et al, J Nanomaterials, 2011, doi.org/10.1155/2011/439162 CuO CTAB sonochemical; Purnachander et al, Energy and Environment FocusVol. 4, pp. 250–255, 2015 CuO CTAB sonochemical; Kim et al Ceramics International 42 (2016) 19454–19460 CuO CTAB sonochemical; Karunakaran, Journal of Alloys and Compounds 580 (2013) 570–577.The papers must be cited in the introduction.

Thank you for the comment and your contribution. The introduction has been extended and the suggestions have been incorporated. Lines 63-80.

-The authors claim that the novelty of this paper is the utilization of NaBH4. However, it is not clear the advantage compared to the classical method (see article reported above)

Thank you for the comment. The introduction has been extended and new paragraphs were incorporated to explain why our method is novelty in comparison with the previous reported ones.

-In general, the quality of the images is poor. For example, in Figure 1 too many spectra are present. and the results are not clear.

The Figure 1 has been modified, and the UV-Vis spectra that was measured every 4 hours were plotted.

-XRD pattern, TEM images and particle distribution of all the prepared materials should be reported in the main text.

All the samples were characterized by UV-Vis spectrophotometry, DLS and Zeta potential. Based on the results of stability over time and lower polydispersity indexes, optimal synthesis conditions were determined. The optimized synthesis was characterized by XRD and TEM.

-The authors should discuss the influence of CTAB and NaBH4 on the final particle size.

Thank you for the comment. The paragraphs that describe the study of the different molar ratios (CTAB:Cu+2 and NaBH4:Cu+2) to optimize the synthesis were explained at the beginning of section 3, before to the characterizations of the optimal conditions and an extensive discussion of the agent effect was added. Lines 163-217

Round  2

Reviewer 1 Report

Authors have addressed most of the concerns of the reviewers improving the quality of the manuscript. However, before its publication there are still some issues that should be improved:

- The use of commas along the manuscript should be revised (e.g. please pay attention to lines 41 and 42 “Zakharova, et al.” and “Alzahrani, et al.” commas after the family name should be removed). 

- Language style in the introduction should be improved. Please revise language e.g. line 149 “as it is already known”. 

- The mechanism has been revised taking into account literature. However, the main novelty of the manuscript is the use of a sonochemical method involving CTAB and NaBH4. Thus, authors should highlight the effect of sonochemical process on the formation of the NPs and their properties, as this is the single difference to previously reported studies.

- In this sense, from Figure 4 is not clear enough the mechanism of formation of the NPs. 

Author Response

POINT TO POINT ANSWER TO QUESTIONS

Referee 1

Authors have addressed most of the concerns of the reviewers improving the quality of the manuscript. However, before its publication there are still some issues that should be improved:

 - The use of commas along the manuscript should be revised (e.g. please pay attention to lines 41 and 42 “Zakharova, et al.” and “Alzahrani, et al.” commas after the family name should be removed).

Thank you for the comment. The commas have been removed.

- Language style in the introduction should be improved. Please revise language e.g. line 149 “as it is already known”.

The paper has been proofread by an English native speaker.

- The mechanism has been revised taking into account literature. However, the main novelty of the manuscript is the use of a sonochemical method involving CTAB and NaBH4. Thus, authors should highlight the effect of sonochemical process on the formation of the NPs and their properties, as this is the single difference to previously reported studies.

Thank you for the comment. A new paragraph has been incorporated (Lines 156-162).

- In this sense, from Figure 4 is not clear enough the mechanism of formation of the NPs.

Figure 4 has been modified. The legend of the figure has been extended and a brief description of each step of the synthesis has been explained. (Lines 304-309)

Reviewer 3 Report

The paper has been significantly improved and can be accepted for publication

Author Response

The reviewer did not make new comments or corrections. He consider that the paper has been significantly improved and can be accepted for publication